# Sensors for Continuous Monitoring of Surgeon’s Cognitive Workload in the Cardiac Operating Room

**DOI:** 10.3390/s20226616

**Published:** 2020-11-19

**Authors:** Lauren R. Kennedy-Metz, Roger D. Dias, Rithy Srey, Geoffrey C. Rance, Cesare Furlanello, Marco A. Zenati

**Affiliations:** 1Division of Cardiac Surgery, Medical Robotics and Computer Assisted Surgery Lab, VA Boston Healthcare System, West Roxbury, MA 02132, USA; marco_zenati@hms.harvard.edu; 2Department of Surgery, Harvard Medical School, Boston, MA 02115, USA; 3STRATUS Center for Medical Simulation, Department of Emergency Medicine, Brigham and Women’s Hospital, Harvard Medical School, Boston, MA 02115, USA; rdias@bwh.harvard.edu; 4Division of Cardiac Surgery, VA Boston Healthcare System, West Roxbury, MA 02132, USA; rithy.srey@va.gov (R.S.); geoffrey.rance@va.gov (G.C.R.); 5HK3 Lab, 20129 Milan, Italy; cesare.furlanello@hk3lab.ai

**Keywords:** cognitive workload, cardiac surgery, heart rate, near-infrared spectroscopy

## Abstract

Monitoring healthcare providers’ cognitive workload during surgical procedures can provide insight into the dynamic changes of mental states that may affect patient clinical outcomes. The role of cognitive factors influencing both technical and non-technical skill are increasingly being recognized, especially as the opportunities to unobtrusively collect accurate and sensitive data are improving. Applying sensors to capture these data in a complex real-world setting such as the cardiac surgery operating room, however, is accompanied by myriad social, physical, and procedural constraints. The goal of this study was to investigate the feasibility of overcoming logistical barriers in order to effectively collect multi-modal psychophysiological inputs via heart rate (HR) and near-infrared spectroscopy (NIRS) acquisition in the real-world setting of the operating room. The surgeon was outfitted with HR and NIRS sensors during aortic valve surgery, and validation analysis was performed to detect the influence of intra-operative events on cardiovascular and prefrontal cortex changes. Signals collected were significantly correlated and noted intra-operative events and subjective self-reports coincided with observable correlations among cardiovascular and cerebral activity across surgical phases. The primary novelty and contribution of this work is in demonstrating the feasibility of collecting continuous sensor data from a surgical team member in a real-world setting.

## 1. Introduction

The potential negative impact of cognitive factors (e.g., cognitive overload) on surgical performance is increasingly being recognized in the literature [1,2,3]. Traditionally, the influence of cognitive factors on preventable adverse events in the operating room (OR) has been largely grounded in theory [1] or driven by investigations of potentially biased post-hoc reports such as morbidity and mortality meetings [2]. There is sparse literature addressing real-time monitoring of cognitive events in the real-world OR setting. Empirical reports utilizing real-time approaches tend to rely on non-invasive sensors to approximate mental states [4]. As sensor technology continues to advance in its accuracy, validity, and usability in experimental settings, surgical data scientists strive to extend its applications to monitor cognitive workload indicators non-invasively in the wild on ultra-sensitive time scales [5,6].

Heart rate variability (HRV) is the most commonly used objective measure of cognitive workload in populations of surgical providers [4]. Inferences derived from HRV analysis extend beyond cardiovascular efficiency, and provide further knowledge of higher-order cognitive processes, according to theories such as the neurovisceral integration model [7] and evidence to support it [8]. Additionally, wearable, wireless heart rate (HR) monitors are capable of detecting states such as mental stress reliably [9]. Beyond being affordable, wireless, non-invasive, and easy to use, the V800 wearable HR monitor manufactured by Polar (Kempele, Finland) in particular has been validated against the traditional electrocardiogram to measure heart rate intervals at rest [10].

In recent years, near-infrared spectroscopy (NIRS) has also been introduced as a valuable modality to assess dynamic neurocognitive changes during various tasks, by providing an estimate of prefrontal cortex (PFC) oxygen saturation via non-invasive sensors affixed to the left and right forehead [11,12]. While wireless NIRS devices have previously been developed for biomedical applications [13], they have yet to be applied to the problem of monitoring prefrontal activity of providers in the OR, and more traditionally apply to patient monitoring approaches.

By combining HRV and NIRS monitoring, a multi-modal approach incorporating both HRV and NIRS sensors simultaneously could effectively characterize the association between the two signals. This has been demonstrated in the literature previously [14], establishing the sensitivity of detecting states of mental overload during simulated flight tasks in an experimental setting. However, the feasibility of collecting both signals in a setting as complex as the cardiovascular operating room has not been previously established in part due to the additional constraints and barriers imposed in the OR setting.

Unlike experimental settings in which sensor data has been previously validated, real-world settings such as the cardiovascular OR present unique physical, procedural, and social/cultural barriers requiring creative solutions, especially when dealing with equipment that is not fully wireless (e.g., wired to a stationary or ambulatory device). Challenges of applying sensors to collect indicators of psychophysiological activity from surgical team members in the OR include physical concerns to rule out the possibility of equipment disrupting the sterile field, creating a hazardous environment by introducing cables, interfering with existing necessary equipment (e.g., head lamp), and limiting the providers’ mobility and flexibility. Specific procedural considerations include the requirement for the attending surgeon and surgeon-in-training to exchange positions during the procedure, requiring them to physically relocate to the opposite side of the operating table, and phases of the surgery requiring the attending surgeon to be seated in order to obtain an optimal field of view. Finally, as with the introduction of any new procedures or equipment, we have to consider social and cultural push-back from clinical providers who may be resistant to adopting change, as well as Hawthorne effect.

Our Medical Robotics and Computer Assisted Surgery (MRCAS) Lab team has previously described the use of HRV to monitor cognitive workload of surgical team members in a real-world setting [15,16,17], while other groups have used functional NIRS (fNIRS) in conjunction with HR during experimental surgical tasks [18,19]. The pilot study reported here is novel in the use for the first time of both HRV and NIRS to simultaneously monitor providers’ cognitive workload during real-world complex surgery. We aimed to assess the feasibility of capturing data from both sensors (HRV and NIRS) equipped to the attending surgeon during an open cardiac surgery procedure.

## 2. Materials and Methods

This research complied with the American Psychological Association Code of Ethics and was approved by the Institutional Review Board at VA Boston Healthcare System and Harvard Medical School (IRB#3047) and was funded by the NIH/NHLBI (PI Zenati).

During a surgical aortic valve replacement (SAVR) procedure characterized by high teaching load and a relatively inexperienced surgical trainee, the attending surgeon was equipped with a wireless heart rate sensor (Polar H10) applied to the chest and linked to a Bluetooth receiver (Polar V800, Kempele, Finland). With this configuration, the surgeon wore the H10 sensors on an elastic, adjustable chest strap, which captured and transmitted all HR data wirelessly to the V800 wristband receiver. Given the constraints of the environment (i.e., sterile field), the wristband was not worn on the wrist, but instead was attached to the waistband to ensure proximity for uninterrupted Bluetooth connectivity. Prior to sensor placement, skin was prepped with alcohol swabs and subsequently dried. Sensor placement was determined according to the manufacturers’ recommended specifications.

The surgeon was also simultaneously equipped with a two-channel cerebral/somatic oximeter (INVOS™ 5100C Cerebral/Somatic Oximeter, Medtronic) applied on the forehead (Figure 1) to collect estimations of left and right PFC regional cerebral oxygen saturation (rSO_2_). In this configuration, each NIRS sensor included one emitter and two photo detectors to capture global activation of the left PFC and global activation of the right PFC. Two depths of light penetration are utilized to subtract out surface data, producing a regional oxygenation value for deeper tissues. rSO_2_ values generated from this device represent the balance of regional oxygen delivery and consumption, as well as any disturbances to this balance. Skin was prepped with alcohol pads and dried prior to sensor placement, and sensor placement was subsequently completed according to manufacturer specifications for adult cerebral sensor placement. According to the recommended specifications, the two sensors were placed directly apposing one another, with sufficient distance between the embedded emitters. One preamplifier connected the disposable NIRS sensors to the INVOS™ monitor via reusable sensor cable connectors.

Relevant clinical characteristics of the monitored surgeon were collected. In particular, the surgeon has well-controlled hypertension, presents with no arrhythmia, is a non-smoker, and drinks one cup of coffee at breakfast daily. Additionally, environmental measures were noted to gauge the general quality of the signal from HRV and NIRS sensors. Ambient temperature and humidity fluctuated minimally, with temperatures maintained between 65 and 69°F over the course of the procedure.

One trained researcher (LKM) was present in the OR during the entire operation to collect ethnographic notes pertaining to relevant surgical phases [20] and events with potential to impose high cognitive load. Examples of intra-operative events recorded during the procedure include a delay for missing equipment, periods of intense teaching activity, arguments with surgical team members, distractions in the environment (e.g., pagers, Vocera badges, etc.), temporal pressures, and difficulties with patient anatomy. Following the case, HR and NIRS data were manually time-synchronized to start at the same second, and mean HR and mean rSO_2_ values were calculated individually for each minute of the procedure. Pre-processed inter-beat interval durations were exported from the Polar platform and artifact detection and removal, as well as mean HR calculations for each minute, were completed using Kubios HRV analysis software [21]. NIRS data were exported from the INVOS™ system, which reported one value representing regional oxygen saturation for each hemisphere roughly every 5 s. Given the paralleled deviations from baseline between the left and right hemispheres, values from each hemisphere were averaged to arrive at one NIRS value for every 5 s. Subsequently, all values within a given minute were averaged to produce one NIRS value for every minute of the procedure.

The total procedure duration from skin incision through skin closure was 2 h and 57 min, which resulted in 177 one-minute samples for each signal. The SAVR procedure was divided into broad a priori surgical phases in reference to the bypass phase, during which the patient’s systemic perfusion is supported by the cardiopulmonary bypass machine via extracorporeal support: (a) pre-bypass, (b) on bypass, and (c) post-bypass. Key surgical phases occurring within these broad bypass phases and intra-operative events annotated during the surgery were superimposed onto a time-series of physiological data. A total of 7 key phases were documented: those occurring pre-bypass included Sternotomy, Heparinization, and Cannulation; those occurring while on bypass included Initiation of Bypass, Aortic Cross Clamp and Cardioplegia Delivery, and Aortotomy and Aortic Valve Replacement; the remaining key phase, Separation from Bypass, occurred primarily after the patient was weaned from extracorporeal support. Additional key phases including Sternal Closure and Post-Operative Debrief, would typically be considered in the broad phase of post-bypass, but were excluded due to missing data during these phases. Similarly, key phases occurring prior to Sternotomy were excluded for the same reason.

## 3. Results

### 3.1. Feasibility of Data Collection

Given the exploratory nature of this case study, use of the INVOS™ 5100C Cerebral/Somatic Oximeter for NIRS data collection was constrained based on the availability of existing clinical equipment at the medical center. Due to its wired connections between the disposable NIRS sensors affixed to the surgeon and the monitor itself, this choice introduced physical and procedural barriers as previously discussed. Figure 2 shows the wired set-up, including the preamplifier, reusable sensor cable connectors, and INVOS™ monitor. Physical barriers required that the preamplifier be positioned within a short distance of the surgeon monitored and that its position be adjusted as the surgeon alternated his location at the operating table. The INVOS™ monitor itself was also placed within a short distance from the preamplifier and remained on the cardiopulmonary bypass pump machine. The cable connecting the preamplifier to the monitor extended a greater distance, allowing the researchers to reposition the preamplifier while keeping the monitor in place.

Due to the wired configuration of the NIRS system, data collection was delayed until the attending surgeon was in the OR continuously and truncated once the attending surgeon concluded his primary operative involvement. While this ensured that there was no data loss during the recording, it also excluded certain phases occurring prior to the surgeons’ entry and continuous involvement (e.g., Anesthesia Induction), as well as phases occurring after the surgeons’ primary surgical involvement had ceased (e.g., Sternal Closure, which is often completed by the trainee, and Post-Operative Debrief).

### 3.2. Preliminary Validation

Kolmogorov–Smirnov tests of normality were completed prior to conducting statistical tests to determine whether HR and NIRS data were normally distributed. A test statistic of 0.044 and *p*-value of 0.868 confirms that HR data followed a normal distribution. Similarly, a test statistic of 0.061 and *p*-value of 0.505 confirms a normal distribution of NIRS data as well.

Given the normal distributions observed in both the HR and NIRS data collected, a Pearson r correlation was calculated. Results of this correlational analysis revealed a moderate but significant positive relationship between mean HR and mean rSO_2_ over the course of the entire surgery, r(177) = 0.67, *p* < 0.001 (Figure 3). Each data point analyzed in this correlation compares the mean HR and mean rSO_2_ value calculated for the same minute.

Start and end times for all high-level bypass phases, as well as the seven key phases, were noted and time-stamped during the observation, allowing for analysis of corresponding physiological values during these phases and sub-phases. Correlations between mean HR and mean rSO_2_ were calculated according the bypass phases described, revealing moderately strong positive correlations during the pre-bypass (r(58) = 0.47, *p* < 0.001) and bypass (r(87) = 0.31, *p* = 0.003) phases of the surgery. The post-bypass phase revealed no relationship between mean HR and mean rSO_2_ values, r(32) = −0.14, *p* = 0.432. Within these broader phases, correlations between data points in sub-phases were also considered, demonstrating a pattern of stronger positive correlations between the signals in earlier sub-phases, compared to predominantly negative associations between signals in later sub-phases (Table 1).

Postoperatively, the attending surgeon subjectively assessed the SAVR procedure as a “moderate-high difficulty teaching case” and cited working with an inexperienced resident as the primary challenge based on a narrative report of events. Specific episodes of high workload and stress (a powerful negative emotion) were also self-reported after the case, including completing the sternotomy despite a technical error of the trainee, cannulating the aorta, and sizing the aortic valve annulus. Self-reported notable events were compared to ethnographic notes, which confirmed the presence of these frustrations through the real-time behavioral observations noted. One additional feature noted in ethnographic observations was the presence of temporal pressure.

Specific key phases in which these notable events occurred are indicated in Table 1. In particular, we noticed that phases with notable events that were characterized by verbally instructing the resident during the Sternotomy phase (during pre-bypass) and temporal pressure during the Aortic Clamp and Cardioplegia phase (during bypass) revealed strong positive and significant correlations between mean HR and mean rSO_2_. Other notable events, including the attending physically taking over for the resident during the Cannulation phase (during pre-bypass) and dealing with unexpected patient anatomy in the Aortotomy phase (during bypass), resulted in moderate and weak negative correlations, respectively, which both approached but did not reach statistical significance.

In addition to correlational analyses, we calculated minimum and maximum differences between data points within each sub-phase to gauge how similar HR and rSO_2_ values were over the course of the sub-phase (Table 2) [22]. Figure 4 provides an exemplified illustration of the relationship between HR and NIRS during the Aortic Clamp and Cardioplegia sub-phase during bypass. This analysis was followed by a two-way Kolmogorov–Smirnov test to statistically evaluate whether distributions from the two data sources differed. Based on these latter analyses, there were no statistically significant differences between the distributions of HR and NIRS data observed across sub-phases (*p* > 0.05 in all sub-phases).

In an effort to approximate cardiovascular autonomic function with more precision, the root mean square of the successive differences (RMSSD), a time-domain measure of HRV [23] was also calculated for each consecutive minute of the surgery. Pearson’s r correlations were calculated to evaluate the relationship between RMSSD and rSO_2_ values during phases, sub-phases, and the entire procedure (Table 3). Due to the reflection of predominantly parasympathetic tone captured by RMSSD, negative correlations between RMSSD and rSO_2_ were expected to correspond to phases and sub-phases with positive correlations between mean HR and rSO_2_. With this in mind, similar trends were observed within certain phases and sub-phases, particularly the Aortic Clamp and Cardioplegia sub-phase.

## 4. Discussion

Observations derived from this case study through preliminary validation efforts demonstrate a significant correlation between mean HR and mean rSO_2_ values during real-life (i.e. not simulated), surgery for the first time in the literature, to the best of our knowledge. Previous work in the healthcare domain has captured both measures simultaneously [18,19], but have done so using simulated surgical tasks, and have failed to discover similar associations or significant correlations between HR and NIRS data. Outside of healthcare, HRV and fNIRS have also successfully demonstrated sensitivities to differing levels of workload, but these findings were in the context of a simulated flight task [14]. Additional preliminary validity suggests a temporal sensitivity of HR and NIRS values in response to ethnographic observations and self-reported stressors. The study reported here represents the first empirical evidence of feasibility and sensitivity in collecting both HR and NIRS data during live surgery.

Increases in intra-operative HR have previously been associated with an elevation in perceived stress as well as elevated salivary cortisol levels (i.e., an objective biomarker of acute stress) [24]. More commonly in healthcare, HR and HRV are utilized as measures reflective of cognitive workload [4]. Similarly, prefrontal activation, detected by NIRS sensors, is known to be associated with cognitive states and load [25]. Specifically, hemodynamic changes in the PFC captured via NIRS sensors has demonstrated changes in cognitive workload during simulated piloting tasks both in isolation [26] and in conjunction with changes in HRV [14].

In contrast to capturing these data in simulated or experimental settings, there are multiple paths forward in terms of using similar approaches in complex real-world settings. Long-term implications of capturing these data in real-world settings include the unique ability to intervene in real time as a means of preventing cognitive overload states. In high-consequence settings, physiological-based interventions such as biofeedback often rely on HRV and are associated with improved performance [27]. Furthermore, the sensitivity of NIRS data affords the opportunity to determine the optimal time to provide notifications or interruptions along the course of a primary task [28], which has otherwise been shown to increase error, time to completion, annoyance, and anxiety [29]. This has similarly been demonstrating using HRV data [30] during real-world cardiac surgeries.

Post-hoc analysis of adverse operative events suggests a substantial influence from cognitive factors on subsequent medical error [2]. Ultimately, a more timely, accurate, and explicit understanding of a surgeons’ cognitive workload fluctuations through psychophysiological monitoring during surgery could inform safety-enhancing cognitive engineering approaches [31].

In summary, in this study an observable relationship was established between real-time manual annotations, subjective reports, and psychophysiological measures collected. In aligning these data sources, we have also preliminarily validated a high level of temporal sensitivity and responsiveness to cognitive workload-induced changes in both HRV and NIRS data. Although we cannot systematically account for influences or biases such as the Hawthorne effect in this study, the triangulation and convergence of multiple data sources provides compelling evidence of validity. These findings lend support for additional studies into the feasibility of systematically collecting multi-modal measures of cognitive workload during surgery through unobtrusive, continuous sensor technology to improve patient safety and team performance.

The hypothesis-generating nature of the study, and of the correlations between signals at various stages of the operative procedure in particular, provides avenues for future work. Based on these preliminary findings, we could conjecture that positive correlations between mean HR and rSO_2_ reflecting a coupling or concordance between the signals may be related to temporal frustrations concerning progress through the case (e.g., multiple verbal corrections impede progress). These moments also appear to be closely related to hindrances that are outside of the control of the surgeon himself. It could be the case that negative, uncoupled changes and a lack of observed relationships are less informative in terms of cognitive state, and more generally reflect difficulties or standard care encountered that the surgeon perceives greater control over (e.g., attending physically taking over). However, additional work is required to understand these relationships more appropriately.

Analysis of the RMSSD component of HRV provided some further insights, offering preliminary evidence to complement the relationship observed between HR and NIRS data in certain sub-phases, but observations of HR were not reliably replicated in RMSSD data. Less information may be gleaned from observing associations between RMSSD, rSO_2_, and notable events in this particular study. Despite prior work demonstrating validity in the Polar V800 device with respect to clinical electrocardiography (ECG) [10], the data acquired in this study were not necessarily representative of a fully restful state. In the case that individual R-peaks were not detected by the V800 sensors, an artificially long inter-beat interval was impossible to correct, given that we did not have a reference point to compare to via a standard ECG device [32]. Therefore, the introduction of occasional motion artifacts when undetected and poorly interpolated by available software presents a limitation in the methodological design of this study. Future work should also consider additional HRV components such as the low frequency: high frequency ratio or percentage of consecutive normal R-peaks differing by at least 50 milliseconds, which can be analyzed using Kubios HRV [21] or open-source physiological signal analysis packages such as *pyphysio* [33].

Due to the nature of this study, we cannot say with certainty that one signal is not anticipating the other, affecting the interpretability of correlational analysis. Future research should seek to model this relationship to investigate the possible prodromic nature of the signals and further to classify the temporal nature of the correlations. Future work should also evaluate these modalities using higher quality research-grade and wireless sensors, which would allow for more sophisticated analyses, 3-dimensional digitization to confirm sensor placement, and more granular discrimination between specific anatomical locations within the PFC. Increased sample sizes would also strengthen the interpretability of observations, as well as standardized approaches to characterizing case difficulty.

## Figures and Tables

**Figure 1 sensors-20-06616-f001:**
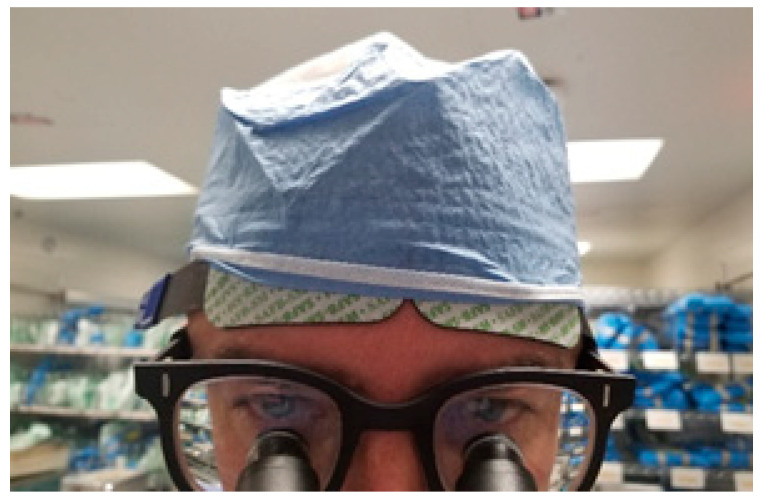
NIRS sensor placement on attending surgeon.

**Figure 2 sensors-20-06616-f002:**
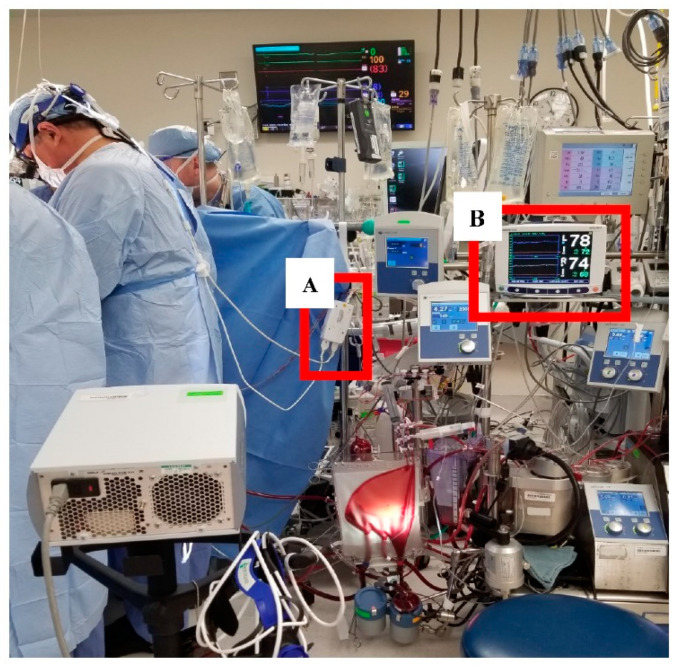
NIRS acquisition device placement in relation to the attending surgeon and cardiopulmonary bypass pump. A. highlights the preamplifier and B. highlights the INVOS™ monitor receiving data from the sensors.

**Figure 3 sensors-20-06616-f003:**
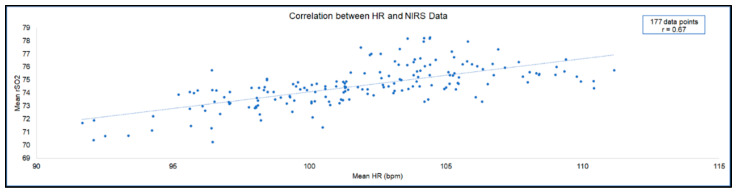
Relationship between mean HR and mean rSO_2_. A significant positive correlation was found between HR and rSO_2_ data. Each data point represents the same 60-second interval of HR data and of rSO_2_ data. The 177 datapoints shown in this figure encompass all phases included in Table 1.

**Figure 4 sensors-20-06616-f004:**
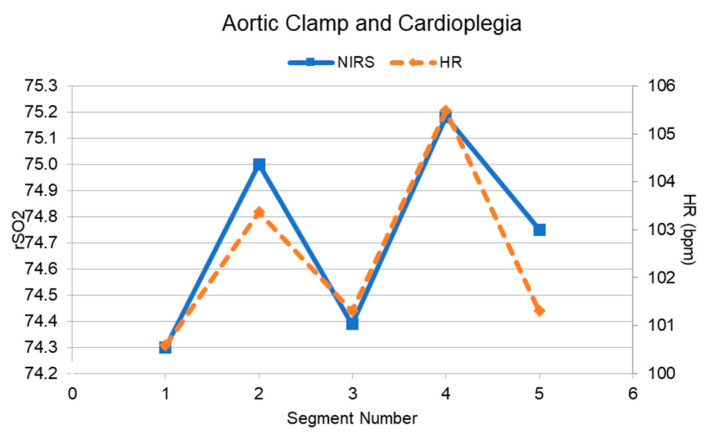
Mean HR and mean rSO_2_ curves during the Aortic Clamp and Cardioplegia sub-phase.

**Table 1 sensors-20-06616-t001:** Pearson’s r correlations between mean HR and mean rSO_2_ values for bypass phases and sub-phases, with notable events observed within sub-phases where applicable. “Other” refers to time points within the corresponding bypass phases, but occurring outside of pre-specified sub-phases.

Bypass Phase	Sub-Phase	Pearson’s r	N	*p*-Value	Notable Events
*1. Pre-bypass*		*0.47*	*58*	*<0.001*	
	1a. Sternotomy	0.58	17	0.014	Resident errors requiring verbal corrections
	1b. Heparinization	0.04	17	0.869	
	1c. Cannulation	−0.53	9	0.142	Resident errors requiring attending to take over
	1d. Other	0.24	15	0.387	
*2. On Bypass*		*0.31*	*87*	*0.003*	
	2a. Initiate Bypass	0.68	4	0.318	
	2b. Aortic Clamp and Cardioplegia	0.91	5	0.031	Temporal pressure (observed)
	2c. Aortotomy	−0.19	66	0.118	Patient anatomy difficulty, irrespective of resident performance
	2d. Other	−0.49	12	0.106	
*3. Post-bypass*		*−0.14*	*32*	*0.432*	
	3a. Separate from Bypass	−0.12	23	0.581	
	3b. Other	0.21	9	0.589	
**Complete case**		**0.67**	**177**	**<0.001**	

**Table 2 sensors-20-06616-t002:** Minimum and maximum differences between HR and NIRS distributions across sub-phases.

Sub-Phase	Minimum Difference	Maximum Difference
1a. Sternotomy	19.97	29.14
1b. Heparinization	21.34	25.17
1c. Cannulation	23.38	28.52
2a. Initiate Bypass	23.20	27.18
2b. Aortic Clamp and Cardioplegia	26.29	30.30
2c. Aortotomy	24.39	35.50
3a. Separate from Bypass	24.72	36.06

**Table 3 sensors-20-06616-t003:** Pearson’s r correlations between RMSSD and mean rSO_2_ values for bypass phases and sub-phases, with notable events observed within sub-phases where applicable. “Other” refers to time points within the corresponding bypass phases, but occurring outside of pre-specified sub-phases.

Bypass Phase	Sub-Phase	Pearson’s r	N	*p*-Value	Notable Events
*1. Pre-bypass*		*0.18*	*58*	*0.185*	
	1a. Sternotomy	−0.17	17	0.497	Resident errors requiring verbal corrections
	1b. Heparinization	0.25	17	0.324	
	1c. Cannulation	−0.02	9	0.968	Resident errors requiring attending to take over
	1d. Other	0.01	15	0.960	
*2. On Bypass*		*−0.06*	*87*	*0.582*	
	2a. Initiate Bypass	−0.26	4	0.740	
	2b. Aortic Clamp and Cardioplegia	−0.99	5	<0.001	Temporal pressure (observed)
	2c. Aortotomy	−0.10	66	0.428	Patient anatomy difficulty, irrespective of resident performance
	2d. Other	−0.31	12	0.333	
*3. Post-bypass*		*−0.18*	*32*	*0.330*	
	3a. Separate from Bypass	−0.26	23	0.230	
	3b. Other	0.06	9	0.882	
**Complete case**		**−0.11**	**177**	**0.151**

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
