# Peer review of "Sensors for Continuous Monitoring of Surgeon’s Cognitive Workload in the Cardiac Operating Room"

_sensors, 2020, doi:10.3390/s20226616_

Round 1
Reviewer 1 Report
The authors describe the concurrent measurement of both HR and NIRS signals in the cardiac surgery operating room. They describe adequately the novelty of their application (first-time concurrent measurement in real-life cardiac surgical setting). The novelty of the work and the constraints that the setting imposed justifies the limited data available (data from one surgeon from one surgery).
There are two points where the authors should provide more information and explanations:
A) In the manuscript the authors investigate the correlation of HR and rSO2. They should also give the curves of HR and rSO2 and statistically compare the HR and rSO2 values in the different phases and sub-phases, in order to relate the findings to previously published material investigating the relation of HR and rSO2 to mental workload and stress.
B) The authors should provide an explanation why some critical stressful phases produced positive correlation between the two metrics (HR and rSO2), while other critical stressful phases did not. This indicates that there could be phases where only one (or even none) of the two metrics provides useful indications about stress monitoring.
Reviewer 2 Report
The paper presents an investigation on the surgical performances, monitoring the surgeon's cardiovascular and cerebral activity during the surgery.
Some statements in the abstract should be re-phrased, to explain more clearly the aim of the paper: "The goal of this study was to investigate the feasibility of overcoming logistical barriers to effectively collect multi-modal psychophysiological inputs through the collection of heart rate (HR) and near-infrared spectroscopy (NIRS) in the wild in the operating room." (line 27-30).
Which is the added medical value of collecting a surgeon's cognitive workload during the operation ? Have the noticed higher cognitive levels influenced the outcome of the surgery ?
The mentioned Hawthorne effect (line 84) does not influence the surgeon's activity and the results of the study ?
Reviewer 3 Report
This is a very interesting, original study with great applicability for understanding autonomic behavior in real-world situations. Some comments are made below in order to better clarify the procedure:
- As this is a pilot study, and the authors reach conclusions about the correlation between NIRS and HR throughout the phases of the aortic valve replacement surgery, it would be interesting to have been informed of the clinical conditions of the studied surgeon, such as age, if he had comorbidities such as hypertension or diabetes, if he smoks or use medications, etc. All of this could interfere with his autonomic state and consequently alter the conclusions of the study. I understand that in this pilot study phase, the main purpose would be to see the feasibility of simultaneously collecting cardiac and brain physiological data, but as the authors also reported conclusions on correlations, this initial assessment of the tested individual becomes relevant.
- The authors mention 7 times throughout the text that the study is Naturalistic. I question this statement, considering that conceptually the naturalistic study is "A type of study in which the researcher very carefully observes and records some behavior or phenomenon, sometimes over a prolonged period, in its natural setting while interfering as little as possible with the subjects or phenomena ". Here in this study, the surgeon knows that he is being observed and the researcher's interference is significant with the placement of the equipment. How to rule out this potential bias effect in the study? Couldn't the Hawthorne effect be occurring, which the authors themselves consider to be a potential cause of bias? I suggest considering the study as an evaluation in "real world situation" instead of naturalistic, as the authors themselves use in other parts of the text (lines 89 and 90): "The pilot study reported here is novel in the use for the first time of both HRV and NIRS to simultaneously monitor providers' cognitive workload during real-world complex surgery ".
- In the Method, on lines 128 and 129, the authors mention that "Pre-processed inter-beat interval durations were exported from the Polar platform and mean HR for each minute was calculated using Kubios HRV analysis software". Kubios HRV is an excellent software for assessing autonomic behavior through heart rate variability. The authors only studied heart rate and not its variability. Why did not include in the analysis at least the most characteristic indexes of HRV assessment such as SDNN, RMSSD, PNN50, LF and HF ?. The simple heart rate could be obtained with any common heart rate monitor. Furthermore, it alone does not represent autonomous behavior as well as the variables mentioned. In addition, the correlation studies with the NIRS could be more illustrative. It is suggested to correlate the NIRS data with some HRV variable (at least RMSSD and HF) and not only with HR.
- The difficulties in installing the NIRS are mentioned in detail, but nothing is said about the Polar frequency meter. Was the watch on the surgeon's wrist? under the apron? Wouldn't that be a reason for high interference? How was the process of filtering potential artifacts done? Do the artifacts captured by Polar also interfere with NIRS registration?
- In line 135 it is mentioned that "The total procedure duration from skin incision through skin closure was 2 hours and 57 minutes, which resulted in 177 one-minute samples for each signal" But at another time (lines 145 to 147) the authors refer that "Additional key phases including Sternal Closure and Post-Operative Debrief, would typically be considered in the broad phase of post bypass, but were excluded due to missing data during these phases. Similarly, key phases occurring prior to Sternotomy were excluded for the same reason ". What was the data loss rate? This is important to know the replicability of the procedure. In addition, it is not clear whether all the 177 points are on the correlation graph. At that time were all phases included?
Round 2
Reviewer 2 Report
The paper presents an investigation on the surgical performances, monitoring the surgeon's cardiovascular and cerebral activity during the surgery.
More technical details should be provided (HRV measurements -SDNN, RMSSD) about the operating surgeon and they could be correlated to the progress of the surgery.
Reviewer 3 Report
Five questions were asked to the authors and their answers were reviewed. In four of them the answers were justified and modifications were made in the text. I believe that these four responses have fully met the expectations. The answer to question 3, however, still does not seem satisfactory to me. The authors replied: "Thank you for this observation! We agree that HRV features would be preferable, but upon initial inspection of HRV data, it was noted that certain features appeared to be less reliable than others, especially those HRV features with high temporal frequency (eg RMSSD pNN50.) When the Polar device did not accurately detect an R-peak in the ECG, and the pre-processing (Polar) and post-processing (Kubios HRV) failed to interpolate a reasonable R-peak in its place, IBIs presented as artificially long durations. Data from the Polar platform is only available in IBI format, after proprietary pre-processing has occurred, eliminating the possibility of manual inspection and correction of the raw ECG. Kubios HRV can be imperfect in interpolating R-peaks Calculating HR data, while a less granular indicator of underlying activity, is less affected by inaccurate R-peak detection and IBI duration, and was preferred in this setting and given this particular dataset. In summary, due to the sporadic and occasional missing R-peaks and inaccurate IBIs in our data, HR was a more appropriate measure of cardiovascular activity than HRV".
I respectfully disagree with that statement. There are several studies validating the use of Polar against the gold standard (Holter test) , with very high correlations in the various variables of Heart Rate Variability [for example: J Strenght Cond Res. 2018 Mar; 32 (3): 716-725 doi: 10.1519 / JSC.0000000000001662; Cilhoroz B et al. (2020) Validation of the Polar V800 heart rate monitor and comparison of artifact correction methods among adults with hypertension. PLoS ONE 15 (10): e0240220. Https://doi.org/10.1371/journal.pone.0240220; Giles, D., Draper, N., Neil, W., 2016. Validity of the Polar V800 heart rate monitor to measure RR intervals at rest. European Journal of Applied Physiology. doi: 10.1007 / s00421-015-3303-9; Vanderlei L, et al. (2008) Comparison of the polar S810i monitor and the ECG for the analysis of heart rate variability in the time and frequency domains. Braz J Med Biol Res 41: 854–859].
In addition, the huge amount of publications in the literature using Heart Rate Variability to study autonomic function, contradict the authors' information that heart rate (HR) is the most appropriate measure of cardiovascular activity than HRV. Therefore, in the impossibility of presentation by the authors, of the comparative values of some of the main variables of HRV, I recommend that, at least, a complementary text be inserted, placing this fact as one of the limitations of the study.
